# A Novel Class of Human ADAM8 Inhibitory Antibodies for Treatment of Triple-Negative Breast Cancer

**DOI:** 10.3390/pharmaceutics16040536

**Published:** 2024-04-13

**Authors:** Nora D. Mineva, Stefania Pianetti, Sonia G. Das, Srimathi Srinivasan, Nicolas M. Billiald, Gail E. Sonenshein

**Affiliations:** 1Department of Developmental, Molecular, and Chemical Biology, Tufts University School of Medicine, Boston, MA 02111, USA; stefania.pianetti@adectopharma.com (S.P.); sonia.das.ay@gmail.com (S.G.D.); srinisrimi@gmail.com (S.S.); nicobilliald@hotmail.fr (N.M.B.); 2Adecto Pharmaceuticals, Inc., Boston, MA 02446, USA

**Keywords:** ADAM8, monoclonal antibodies, inhibitory mAbs, triple negative, breast cancer, targeted treatment

## Abstract

New targeted treatments are urgently needed to improve triple-negative breast cancer (TNBC) patient survival. Previously, we identified the cell surface protein A Disintegrin And Metalloprotease 8 (ADAM8) as a driver of TNBC tumor growth and spread via its metalloproteinase and disintegrin (MP and DI) domains. In proof-of-concept studies, we demonstrated that a monoclonal antibody (mAb) that simultaneously inhibits both domains represents a promising therapeutic approach. Here, we screened a hybridoma library using a multistep selection strategy, including flow cytometry for Ab binding to native conformation protein and in vitro cell-based functional assays to isolate a novel panel of highly specific human ADAM8 dual MP and DI inhibitory mAbs, called ADPs. The screening of four top candidates for in vivo anti-cancer activity in an orthotopic MDA-MB-231 TNBC model of ADAM8-driven primary growth identified two lead mAbs, ADP2 and ADP13. Flow cytometry, hydrogen/deuterium exchange–mass spectrometry (HDX-MS) and alanine (ALA) scanning mutagenesis revealed that dual MP and DI inhibition was mediated via binding to the DI. Further testing in mice showed ADP2 and ADP13 reduce aggressive TNBC characteristics, including locoregional regrowth and metastasis, and improve survival, demonstrating strong therapeutic potential. The continued development of these mAbs into an ADAM8-targeted therapy could revolutionize TNBC treatment.

## 1. Introduction

Triple-negative breast cancer (TNBC) is defined by a lack of expression of the hormone receptors estrogen receptor α (ER) and progesterone receptor (PR) and of the human epidermal growth factor receptor 2 (HER2). Accordingly, TNBC patients cannot benefit from endocrine- and HER2-targeted therapies that are the standard of care (SoC) for most other breast cancer patients [1]. TNBC accounts for 10–15% of breast cancer cases and occurs preferentially in young, pre-menopausal women (<40 years of age) and in women of African American descent (American Cancer Society). Historically, the TNBC SoC has consisted of surgery, radiation, and chemotherapy (CT), which are not curative in ~30% of patients. In these cases, disease recurs within 1–3 years, presenting as locoregional relapse and/or distant metastases to the bones, lungs, pleura, brain, or liver, and progresses rapidly to death [2,3]. Due to this aggressive clinical course and the lack of targeted treatment options, TNBC has a higher risk of death within 5 years of the initial diagnosis compared to other breast cancer subtypes [2]. Two recent additions to the TNBC therapeutic armamentarium, the immunotherapy pembrolizumab (Keytruda, Merck & Co Inc., Rahway, NJ, USA) and the antibody (Ab)–drug conjugate sacituzumab govitecan-hziy (Trodelvy, Gilead, Foster City, CA, USA), offer new hope for patients; however, response rates and duration are quite limited [4,5]. Thus, novel treatment options are urgently needed to improve patient outcomes. Using large-scale transcriptional analyses, we identified the gene expressing the cell surface A Disintegrin And Metalloprotease 8 (ADAM8) protein as one of the most overexpressed in breast cancer compared to normal breast tissue and an independent predictor of poor disease-free (DFS) and overall (OS) survival [6]. Immunohistochemistry of primary TNBC tumors revealed that 34% (17/50) were ADAM8-positive (ADAM8+) [6]; this observation was recently confirmed in a second data set (22/61 = 36%; [7]). In an orthotopic mammary fat pad (MFP) mouse model, MDA-MB-231 TNBC cells with stable ADAM8 knockdown formed small tumors, which failed to induce angiogenesis and grow beyond a palpable size, shed fewer circulating tumor cells (CTCs) into the bloodstream, and resulted in profoundly reduced metastasis compared to control ADAM8-expressing MDA-MB-231 cells [6,8]. Replicating this effect with an ADAM8 pharmacologic inhibitor could provide significant benefit to a large section of the TNBC population that is underserved by current therapies. 

Human ADAM8 is synthesized as an inactive 120 kDa [824 amino acid (AA)] proform with a signal peptide (AA1-16) and an inhibitory amino-terminal prodomain (PRO, AA17-191). These features are followed by a metalloproteinase domain (MP, AA192-406), a disintegrin domain (DI, AA407-496), a cysteine-rich domain (CRD, AA497-612), and an EGF-like domain (ELD, AA613-640). A transmembrane region (TM, AA641-678) and an internal cytoplasmic tail (CYTO, AA679-824) complete the full protein [9]. Upon dimerization or multimerization, ADAM8 autocatalytically cleaves its PRO, leading to the formation of a 90 kDa membrane-anchored active form, with functional MP and DI [10]. The further removal of the MP can result in a 60 kDa cell surface remnant form with DI activity. Our mechanistic studies in breast cancer cells demonstrate that ADAM8 has no direct effect on cell proliferation or survival, but rather on the tumor cell surface and microenvironment [6]. The MP releases various factors (e.g., VEGF-A, PDGF-AA, and angiogenin) at the tumor cell surface that can promote peritumoral angiogenesis and thereby tumor growth. Furthermore, the DI activates β1-Integrin signaling, which facilitates breast cancer cell migration, interaction with and adhesion to endothelial cells, entry into the bloodstream, and metastasis [6,11].

To date, there have been several attempts to develop ADAM8 inhibitors for therapy. Some focused on the use of small molecules directed exclusively against the ADAM8 MP but failed to generate inhibitors with sufficient protein specificity due to enzymatic pocket similarities with other ADAM family members [12,13]. Others have concentrated on cyclic peptides that block ADAM8 multimerization and therefore protein activation. These agents have so far failed to move forward in development due to an extremely short half-life [14,15]. We hypothesized that a monoclonal Ab (mAb) that simultaneously inactivates the MP and DI activities of ADAM8 would provide both the specificity and efficacy needed for a successful future therapy. Using commercially available reagent-grade anti-ADAM8 Abs, we showed proof-of-concept that dual MP and DI inhibitory Abs can be isolated, and using one such inhibitor demonstrated efficacy against both primary tumor growth and metastasis; the mechanism driving dual inhibition was not elucidated [6]. Because reagent-grade Abs cannot be translated into the clinic [e.g., due to a lack of characterization and specificity], here using the hybridoma method followed by an extensive cell-based screening approach, we isolated a panel of novel dual MP and DI inhibitory mAbs (termed ADPs) that bind ADAM8 with high affinity and specificity. In vivo mouse screening and epitope mapping identified ADPs with strong anti-cancer activity that improved survival via binding to sequences within the DI. Overall, our studies identify a mechanism of action for dual inhibitory ADAM8 mAbs and two promising lead ADPs for the development of a future TNBC intervention.

## 2. Materials and Methods

### 2.1. Cell Lines and Culture Conditions

Hybridomas producing ADP Abs were generated at ChemPartner (Shanghai, China) as described below. Isotype-matched control mAb producing hybridomas [BrdU IgG1 clone G3G4 (RRID: AB_2314035) and anti-Manduca sexta ecdysone IgG2b clone 10F1 (RRID: AB_528210)] were purchased from the Developmental Studies Hybridoma Bank (DSHB) [Iowa City, IA, USA]. All hybridomas were grown in HyClone CCM1 medium (GE Healthcare, Chicago, IL, USA) and each batch confirmed mycoplasma-free using a PCR-based test (MP0025, VenorGeM Mycoplasma Detection Kit, MilliporeSigma, Burlington, MA, USA). Human embryonic kidney cells HEK293 [RRID: CVCL_0045] were purchased from American Type Culture Collection (ATCC, Manassas, VA, USA) and maintained in DMEM supplemented with 10% Fetal Bovine Serum (FBS). HEK293 cells expressing full-length human ADAM8 [HEK-A8], remnant human ADAM8 (HEK-REM), or control empty vector (EV) pCDNA3.1 [HEK-EV] were prepared as reported [11] and maintained stably in 500 μg/mL Geneticin (G5005, Teknova, Hollister, CA, USA). Chinese Hamster Ovary (CHO) cells [RRID: CVCL_0213] expressing α9β1-Integrin were generated and maintained as reported previously [16] [kindly provided by Yoshikazu Takada (UC Davis School of Medicine, Sacramento, CA, USA)]. HEK293 cells transiently expressing ADAM33 were generated following transient transfection [48 h] with an ADAM33 construct (HsCD00419548, PlasmID Repository, Dana-Farber/Harvard Cancer Center DNA Resource Core, Boston, MA, USA) [HEK-A33] or EV DNA (25890, Addgene, Watertown, MA, USA) [HEK-EV_2_] using Lipofectamine 2000 (11668019, Thermo Fisher Scientific, Waltham, MA, USA) according to the manufacturer’s protocol. Human TNBC MDA-MB-231 cells (RRID: CVCL_0062) tagged with luciferase [MDA-MB-231-luc] were kindly provided by Charlotte Kuperwasser (Tufts University School of Medicine, Boston, MA, USA) and maintained as recommended by ATCC and in the presence of 500 μg/mL Geneticin (G5005, Teknova). All cells used in the laboratory were confirmed mycoplasma-free as above. All human cell lines were authenticated within the past three years using short tandem repeat analysis (Labcorp, Cincinnati, OH, USA) and demonstrated 100% identity with appropriate ATCC lines.

### 2.2. ADP Generation

Anti-ADAM8 Abs were made by the hybridoma method [17] using recombinant human ADAM8 [rHuADAM8] (AD8-H5223, ACRO Biosystems, Newark, DE, USA) injection into Balb/c and SJL mice and fusion of B cells to Sp2/0-Ag14 (RRID: CVCL_2199) myeloma cells. Hybridoma supernatants were screened for clones producing Abs with highly specific ADAM8-binding and dual MP and DI inhibitory activity, as described below. Following subcloning by serial dilution, 18 clones making such mAbs (termed ADPs) were isolated. Abs were purified from hybridoma supernatants using protein A columns. Sterile filtered, purified Abs in phosphate-buffered saline (PBS) had low endotoxin (<2 EU/mg) and >95% purity (SDS-PAGE). Isotype and light chain type were determined using an SBA Clonotyping System-HRP kit (5300-05, Southern Biotech, Birmingham, AL, USA).

### 2.3. ELISA

Briefly, 96-well plates, coated with 1 μg/mL rHuADAM8 or recombinant human ADAM9 (939-AD-020, R&D Systems, Minneapolis, MN, USA), ADAM12 (AD2-H5228, ACRO Biosystems), or ADAM15 (10517-H08H, Sino Biological, Houston, TX, USA) were exposed to sera, supernatants, or purified ADPs. For serum, 6 dilutions (1:100–1:10,000,000) of pre-bleed and test bleed samples were used. For hybridomas, 50 µL of supernatant was tested. For purified ADPs, 8 dilutions (10^−5^–10^3^ nM) were used. Test bleeds (1:1000) or FBS (1:100) and CCM1 or normal mouse IgG (1 µg/mL) were used as positive and negative controls, respectively. Goat anti-mouse IgG (Fc specific)-HRP (1:5000, A0168, MilliporeSigma, RRID: AB_257867) was used as secondary Ab. Signal detection was with 1-Step Ultra TMB-ELISA Substrate Solution (34028, Thermo Fisher Scientific).

### 2.4. Flow Cytometry

For ADP binding to ADAM8, 3000 HEK-A8, HEK-REM, or HEK-EV cells in flow cytometry buffer (1% BSA, 0.1% sodium azide in PBS) were exposed to sera (1:100), supernatants (10 μL), or ADPs (0.1–10 μg/mL). As negative and positive controls, 1 μg/mL normal mouse IgG and goat anti-mouse ADAM8 Ab (AF1031, R&D Systems, RRID: AB_354549), respectively, were analyzed. Alexa Fluor 488 donkey anti-mouse IgG (1:1000, A-21202, Thermo Fisher Scientific, RRID: AB_141607) or Alexa Fluor 488 donkey anti-goat IgG (1:1000, A-11055, Thermo Fisher Scientific, RRID: AB_2534102) were used as secondary Abs. For ADP cross-reactivity to ADAM33, which is primarily cytoplasmic, 10^6^ HEK-A33 or HEK-EV_2_ cells were fixed in cold 4% paraformaldehyde and permeabilized with 0.1% Saponin before exposure to ADP or control IgGs (2 μg/50 μL sample). An ADAM33 Ab (LS-C124915, LifeSpan Biosciences, Seattle, WA, USA, RRID: AB_10831502) [2 μg/50 μL sample] and an Alexa Fluor 488 chicken anti-mouse IgG (A-21200, Thermo Fisher Scientific, RRID: AB_2535786) [1.25 μg/50 μL sample] were used as positive control and secondary Ab, respectively. All samples were run on a BD FACSCalibur (BD Biosciences, Woburn, MA, USA).

### 2.5. Biacore Surface Plasmon Resonance

To evaluate binding to rHuADAM8, each ADP was used as ligand in a multiple cycle kinetics study on a Biacore T200 surface plasmon resonance system (GE Healthcare). ADPs were captured using anti-mouse Fc IgG attached to a dextran matrix, and rHuADAM8 was added as analyte at concentrations ranging from 3.75 nM to 200 nM. HBS-EP+ running buffer (10 mM HEPES, 0.15 M NaCl, 3 mM EDTA, and 0.05% Surfactant P20) was passed at a flow rate of 30 µL/min and antigen injected at 30 µL/min. Association and dissociation phases were carried out for 180 and 600 seconds (s), respectively. Surface regeneration was performed for 30 s at 30 µL/min 10 mM Glycine pH 1.5. Association rate (k_a_), dissociation rate (k_d_), and equilibrium dissociation (K_D_) constants were calculated via the T200 Software.

### 2.6. ADAM8 Functional Assays

To assess mAb ability to inhibit MP activity, a cell-based CD23 cleavage assay was performed, as described previously [6], in the presence of ADPs either concentrated from hybridoma supernatants or purified. Supernatants were concentrated (10×) with Amicon Ultra Centrifugal Filters (MilliporeSigma), dialyzed against PBS using Micro Float-A-Lyzer dialysis units (Spectrum Laboratories, Rancho Dominguez, CA, USA), and quantified with an Easy Titer IgG Assay kit (Thermo Fisher Scientific). Isotype-matched controls were IgG1 (clone G3G4), IgG2b (clone 10F1), and IgG2c (clone 6.3, ASB-1220, Nordic Biosite, Wayne, PA, USA). Anti-ADAM8 MAB1031 Ab (R&D Systems, RRID: AB_2305036) was used as positive control for MP inhibition. To assess the ability of ADPs to inhibit DI activity, we measured their effects on the adhesion of CHO cells expressing α9β1-Integrin to rHuADAM8, as reported previously [16], using either dialyzed or purified ADPs. Control IgGs and MAB1031 were used as negative and positive controls for DI inhibition, respectively.

### 2.7. Epitope Binning

ADPs were clustered into epitope groups by competitive ELISA. Briefly, purified ADPs (1 µg/mL) were individually coated on 96-well plates. Plates were exposed for 1 h to biotinylated rHuADAM8 that was pre-incubated with excess of a second competitor ADP (ADPC) or control IgG. Streptavidin-HRP (1:5000) was used for detection. Signal was developed using 1-Step Ultra TMB-ELISA Substrate Solution (34028, Thermo Fisher Scientific). Extent of competition between 2 ADPs was calculated as (1-OD450 ADPC/OD450 control IgG)%. Cross-competition of 75% or more was used to delineate epitope clusters.

### 2.8. Hydrogen/Deuterium Exchange–Mass Spectrometry (HDX-MS)

HDX-MS with pepsin/protease XIII digestion was performed by NovaBioAssays (Woburn, MA, USA) on an ultra-performance liquid chromatography–mass spectrometry (LC-MS) system composed of a Waters Acquity UPLC coupled to a Q Exactive plus Hybrid Quadrupole-Orbitrap Mass Spectrometer (Thermo Fisher Scientific). Initial pepsin/protease XIII digestion of rHuADAM8 (AA1-497, 1031-AD-020, R&D Systems), LC-MS, and MS/MS data search against human ADAM8 in Mascot revealed 69.7–74% protein coverage, consistent with autocatalytic removal of the PRO. Next, rHuADAM8 and ADP2, ADP3, or ADP13 (8 µg: 24 µg) were incubated with deuterium oxide for 0, 60, 600, or 3600 s. Following pepsin/protease XIII digestion, LC-MS was performed, as above. Mass spectra were recorded in MS-only mode and processed using HDX WorkBench software [18]. Deuterium levels at the various peptides were monitored from the mass shift on LC-MS vs. native form (peptides at time zero before deuterium addition). Deuterium buildup curves vs. exchange time were plotted and ADAM8 sequences with reduced deuterium uptake indicative of ADP binding identified. 

### 2.9. Mutagenesis Mapping

To identify AAs in ADAM8 mediating ADP2 or ADP13 binding, alanine (ALA) scanning mutagenesis and flow cytometry were performed by Integral Molecular (Philadelphia, PA, USA), as published [19]. Briefly, a mutation library of human ADAM8 was created by high-throughput, site-directed mutagenesis. Each residue in the MP and DI (AA192-AA497) was individually mutated to ALA and, where ALA was the original AA, to serine (SER). This library was transiently transfected into HEK293 cells and expression confirmed using flow cytometry with an anti-ADAM8 Ab (Control A8 Ab, MAB10311, R&D Systems, RRID: AB_2273524), whose binding [within the cysteine-rich domain and EGF-like domain (CRD-ELD) region of ADAM8] is not affected by the mutations. HEK293 cells expressing wild-type (WT) ADAM8 or EV DNA were used as positive and negative controls, respectively. Following transfection, clones were incubated with antigen-binding fragments (Fabs) of ADP2 (0.50 μg/mL) and ADP13 (5.00 μg/mL), or Control A8 Ab (0.16 μg/mL). Primary Abs were detected using Alexa Fluor 488 F(ab′)2 specific goat anti-mouse IgG (1:200, 115-546-006, Jackson ImmunoResearch Laboratories, West Grove, PA, USA, RRID: AB_2338860) for ADP Fabs and Alexa Fluor 488 goat anti-mouse IgG (1:400, 115-545-003, Jackson ImmunoResearch Laboratories, RRID: AB_2338840) for Control A8 Ab. Mean cellular fluorescence was determined using an Intellicyt iQue flow cytometry platform (Sartorius, Ann Arbor, MI, USA). Background fluorescence was subtracted, and Ab reactivity normalized to WT ADAM8.

### 2.10. Orthotopic TNBC Mouse Models

#### 2.10.1. Primary Tumor Growth

Female NOD/SCID mice (10 weeks old) were implanted with 0.5 × 10^6^ MDA-MB-231-luc cells in the 4th inguinal MFP, as reported [6]. When tumors reached ~50–75 mm^3^, mice were treated by intraperitoneal (i.p.) injection 3×/week with 0–30 mg/kg ADP vs. IgG1 (clone G3G4) or IgG2b (clone 10F1). Tumor size was measured with calipers and volume calculated as (Length × Width^2^)/2. Mice were sacrificed when the average of the IgG group was ~1 cm^3^.

#### 2.10.2. Tumor Regrowth

Nanoparticle Albumin-Bound Paclitaxel (NPAC) [Abraxane, Celgene, Summit, NJ, USA] was obtained from the Tufts Medical Center Pharmacy (Boston, MA, USA). To determine an effective dose, female NOD/SCID mice bearing MDA-MB-231-luc cell MFP tumors (derived as above) were treated with 1 cycle of 5 consecutive daily intravenous (i.v.) injections of 0–30 mg/kg NPAC in vehicle Saline. The maximum effective dose with no substantial adverse effects was 10 mg/kg. Two cycles, with one week of rest in between, were selected to achieve tumor regression, as previously reported [20]. Next, female NOD/SCID mice bearing ~150 mm^3^ MDA-MB-231-luc cell-derived MFP tumors (Day 19 post-implantation) were divided into 4 groups: (a) IgG + Saline, (b) ADP + Saline, (c) IgG + NPAC, and (d) ADP + NPAC. NPAC or Saline i.v. treatment was initiated on Day 20. ADP or IgG was administered by i.p. injection, starting with a loading dose of 20 mg/kg, followed by maintenance doses of 10 mg/kg 3×/week. Pharmacokinetic studies determined this regimen achieves steady-state Ab concentrations in the blood of mice. ADP treatment was started with NPAC and continued throughout the time course; endpoint for evaluation was a tumor volume of ~1 cm^3^.

#### 2.10.3. Metastasis and Outcome

A neoadjuvant treatment protocol was performed, as published [6]. Briefly, ~2 weeks after MFP cancer cell implantation, female NOD/SCID mice carrying 50–75 mm^3^ tumors derived from MDA-MB-231-luc cells, which preferentially metastasize to the bone, were treated by i.p. injection 3×/week with 10 mg/kg ADP or control IgG. When tumors reached a volume of ~200 mm^3^ in the control group (~1 week after treatment initiation), tumors were surgically removed, and ADP or IgG treatment continued for another 12.5 weeks. Primary site was assessed for recurrence using palpation 3×/week. Mice were sacrificed when recurrent tumors reached ~0.9 cm^3^ or at the end of the experiment (12.5 weeks/Day 88 post-resection). This experimental endpoint was selected to ensure the NOD/SCID mice used did not surpass 26 weeks of age at which point, based on our observations, this strain can develop spontaneous lymphomas. Dissected bones (skull, spine, and front and hind legs) were assessed for metastasis using Biophotonic imaging of luciferase activity from the tagged cancer cells on a Xenogen IVIS-200 machine (PerkinElmer, Waltham, MA, USA).

### 2.11. Statistical Analyses

Tumor volume comparisons between two treatment groups were performed using the two-tailed Student’s *t*-test. Kaplan–Meier curves for DFS and OS were generated using GraphPad Prism 10 software and statistical analyses performed using the Log rank test. Dissected bone luciferase signal comparisons to evaluate extent of metastasis between treatment groups were carried out using unpaired *t*-tests in GraphPad Prism. *p* < 0.05 was considered statistically significant.

## 3. Results

### 3.1. Development of a Panel of ADAM8 MP and DI Inhibitory mAbs

A traditional hybridoma method combined with an extensive screening protocol was used to develop ADAM8 MP- and DI-neutralizing mAbs. Purified rHuADAM8 (AA17-497) was injected into both Balb/c and SJL mice to generate a broad range of immune responses. The cell supernatants from the resulting hybridomas were screened by flow cytometry with HEK-A8 cells, which express cell surface, native conformation human ADAM8, and by ELISA with the rHuADAM8 immunogen to confirm that isolated clones made mAbs with high ADAM8-binding activity under both conditions. Hybridomas producing mAbs that cross-reacted with human ADAM9, ADAM12, or ADAM15, which are homologous to ADAM8, were identified in ELISA and excluded. The supernatants from the remaining hybridomas were tested for ADAM8 inhibitory activity using cell-based MP and DI domain functional assays. Using this strategy, eighteen hybridomas were isolated and subcloned and their purified mAbs isotyped to confirm single clone origin. These novel anti-ADAM8 mAbs were either IgG1 or IgG2 (all κ L chain) and termed ADPs. They specifically bound both native (Table 1) and recombinant (Table 2) ADAM8, had low K_D_s [1.3 × 10^−9^ M to 7.23 × 10^−8^ M] (Table 2), and showed no cross-reactivity to the related ADAM9, ADAM12, and ADAM15 proteins (Appendix A).

Epitope binning performed using competitive ELISA determined that ADPs bind to five epitope clusters on ADAM8 (Appendix A and Figure 1). Epitopes 2, 3, 4, and 5 overlapped partially, while Epitope 1 was unique (non-overlapping). Two clusters (Epitopes 1 and 3) contained the most ADPs, indicating a high level of conformational accessibility and immunogenicity (Figure 1).

Nine ADPs with the highest ADAM8-binding activities (K_D_ = 1.3 × 10^−9^–9.75 × 10^−9^ M) and/or of the more abundant epitope groups were selected for a comparison of dual MP and DI inhibitory activity in cell-based assays vs. the research-grade MAB1031 Ab identified as a prototype reagent in our proof-of-concept studies [6]. Specifically, they were screened for their ability to inhibit (1) MP activity and (2) DI activity, as described in Figure 2. Eight of these ADPs, which bound Epitopes 1, 2, and 3, had potent MP and DI inhibitory activity, i.e., comparable to or better than MAB1031 with respect to MP (Figure 2A), DI (Figure 2B), or both activities. ADP4 displayed MP inhibitory activity but had minimal DI inhibitory activity, possibly as a result of the loss of the original clone during subcloning or cell banking, and was eliminated from further studies.

Overall, 8 anti-human ADAM8 mAbs with high affinity and specificity for native ADAM8 and potent in vitro dual MP and DI inhibitory activity were identified.

### 3.2. ADP2, ADP13, and to a Lesser Extent ADP3 Inhibit Primary TNBC Tumor Growth

Potent in vitro activity does not necessarily translate into in vivo therapeutic efficacy. Thus, ADPs were next tested in a single-dose pre-existing orthotopic ADAM8+ MDA-MB-231 TNBC cell line-derived primary tumor growth model. Four ADPs representative of Epitope groups 1 [ADP19 K_D_ = 9.1 × 10^−9^], 2 [ADP13 K_D_ = 1.3 × 10^−9^], and 3 [ADP2 K_D_ = 3.3 × 10^−9^ and ADP3 K_D_ = 1.8 × 10^−8^] were selected. A preliminary study using ADP13 indicated that the dosing of female NOD/SCID mice with 10 mg/kg 3×/week resulted in significant tumor growth inhibition (TGI). Treatment with 10 mg/kg of the selected ADPs vs. control isotype-matched IgGs was initiated at a tumor volume of ~50–75 mm^3^ and continued 3×/week until the control group approached a volume of 1 cm^3^ (the limit of our IACUC protocol) (Figure 3). The TGI was 52% with Epitope 2 Ab ADP13 (*p* = 0.0089), and 47% and 28% with Epitope 3 Abs ADP2 (*p* = 0.0001) and ADP3 (*p* = 0.0155), respectively (Figure 3A). Epitope 1 Ab ADP19 did not sustain TGI, despite a low K_D_ and potent in vitro dual inhibitory activity (Figure 3A). Subsequent dose–response curves using 1, 3, 10, or 30 mg/kg ADP2 or ADP13 vs. their control IgGs confirmed that TGI occurred in a dose-dependent manner and determined 10 mg/kg was the maximal effective dose (Figure 3B,C). Thus, Epitope 2 and 3 binding Abs have substantial in vivo anti-cancer activity.

### 3.3. Interactions of ADP2, ADP3, and ADP13 with ADAM8 Map to the DI

To begin to map the binding of the active ADPs ADP2, ADP3, and ADP13 to ADAM8, flow cytometry was performed using either HEK-A8 cells that express a full-length ADAM8 vector, which generates the proform and derived active and remnant forms, or HEK-REM cells that express a deletion construct lacking the PRO and MP, which generates only the remnant form (Figure 4). ADP2, ADP3, and ADP13 bound to the surface of both the HEK-A8 and HEK-REM cells (Figure 4A), indicating that Epitopes 2 and 3 were within AA407 to AA640 of ADAM8, containing the extracellular DI, CRD, and ELD shared by the full-length and remnant constructs (Figure 4B). Similar testing of additional Epitope 3 Abs (ADP1 and ADP12) confirmed this finding (Appendix A). Given that the immunogen (Figure 4B) used to generate these Abs contains only the PRO, MP, and DI, the results suggested that Epitope 2 and 3 Abs were in fact binding within the common DI sequence (Figure 4B). 

Epitope mapping at the peptide level was next performed using HDX-MS of rHuADAM8. For this analysis, rHuADAM8 was incubated with deuterium oxide in the absence or presence of ADP2, ADP3, or ADP13 and then subjected to pepsin/protease XIII digestion. The inhibitory effect of Ab binding on the amount of deuterium in the resulting peptides was determined using LC-MS. Consistent with the flow cytometry data, the three ADPs showed a specific reduction in the deuterium uptake at sequences within the DI (Figure 4C). These sequences, which spanned AA423 to AA491, were partially overlapping between ADP3 and ADP2 and between ADP2 and ADP13 and were located within the putative hinge of the C-shaped structure that defines the tight 3D form of the ADAM8 ectodomain (Figure 4C and Appendix A) [21]. These findings were also consistent with the binning described in Figure 1 and Appendix A, which showed the epitopes for (i) ADP3 and ADP2 and (ii) ADP2 and ADP13 substantially overlapped (~90%), whereas there was no significant epitope overlap between ADP3 and ADP13. 

To identify the specific AAs that mediate the interaction of the top inhibitors ADP2 and ADP13 with ADAM8, shotgun ALA scanning mutagenesis was combined with high-throughput flow cytometry. For this analysis, a library of human ADAM8 expression constructs was generated with single ALA mutations introduced into each AA within the MP and DI; when ALA was the original AA, it was mutated to SER. Following transfection into HEK293 cells, the library was screened by flow cytometry using Fabs of ADP2 and ADP13, or a positive control ADAM8 Ab (Control A8 Ab), whose binding was outside the MP and DI region and therefore not affected by the mutations. Mutated residues were identified as being critical to the ADP2 or ADP13 epitope if they supported the reactivity of the Control A8 Ab but not that of the test Fab. This strategy facilitated the exclusion of mutants that are locally misfolded or have expression defects. One AA for ADP2 (E444) and four for ADP13 (G445, Q447, K458, and R482) reached the criteria for critical binding residues (i.e., the mutation maintained Control A8 Ab binding at >70% of that seen with WT ADAM8 but reduced test Ab binding to <20% of WT) (Figure 5A). Three AAs for ADP2 (R431, G445, and K458) and two for ADP13 (V459 and A462) were residues of secondary importance, i.e., that did not reach the <20% of the WT binding criterion for critical residues, but still led to a substantial reduction in Fab binding (Figure 5A). In combination with their proximity to critical residues, these findings indicated that they are part of the Ab epitope.

Overall, there was tremendous concordance between the AAs identified by shotgun mutagenesis and HDX-MS, i.e., all the critical AAs mapped by mutagenesis fall within the peptide regions mapped by HDX-MS within the DI domain. The positions of the ADP2 and ADP13 critical and secondary binding residues, within the C-shaped structure of the ADAM8 ectodomain, are indicated in an ADAM8 crystal structure model based on the structure of vascular apoptosis-inducing protein-1 (PDB: 2ERP, [21]) (Figure 5B,C). Lastly, as this specific region of ADAM8 bore some similarity to sequences within ADAM33, we confirmed ADP2 and ADP13 lacked cross-reactivity to ADAM33 using flow cytometry analysis of HEK293 cells overexpressing this protein (Appendix A). Together, these results indicate that ADP2, ADP3, and ADP13 define a novel family of ADAM8 mAbs whose activity is mediated via selective binding to the DI.

### 3.4. ADP2 and ADP13 Inhibit TNBC Tumor Regrowth

Based on superior in vivo activity (Figure 3), ADP2 and ADP13 were prioritized for further preclinical screening in orthotopic mouse models that reflect the development of locoregional relapse or distant metastases, which lead to the high mortality rate seen in TNBC patients. To assess their effect on tumor regrowth, ADP2 and ADP13 were tested with the TNBC SoC CT NPAC—a formulation of Paclitaxel—which was selected based on its greater stability and clinical efficacy compared to either Paclitaxel or Docetaxel [22,23]. NOD/SCID mice bearing advanced, rapidly growing MDA-MB-231-luc cell-derived mammary tumors were treated with either (a) IgG + Saline, (b) ADP2 or ADP13 + Saline, (c) IgG + NPAC, or (d) ADP2 or ADP13 + NPAC (Figure 6). NPAC was given in two cycles with one week of rest in between; each cycle consisted of 5 consecutive daily i.v. injections of 10 mg/kg NPAC. An equivalent volume of vehicle Saline was given to the control animals. ADP2, ADP13, or the respective IgG (10 mg/kg) was given by i.p. injection 3×/week throughout the time course, starting with the first CT cycle. The endpoint for the evaluation of TGI was an average tumor volume approaching 1 cm^3^ in the appropriate control groups.

The IgG + Saline-treated tumors grew rapidly, reaching 1 cm^3^ on Day 31 as expected (Figure 6A,B). Treatment with either ADP2 or ADP13 as monotherapy led to a substantial reduction in tumor volume [ADP2 TGI = 38%, *p* = 0.011; ADP13 TGI = 22%, *p* = 0.060] (Figure 6A,B). This inhibition was more moderate than previously seen (Figure 3) as a tumor regrowth model requires much more advanced disease (3× larger tumor volume at start of therapy) to ensure tumors are able to recur following an initial CT-mediated regression. The NPAC treatment led to dramatic disease regression such that the tumors were barely palpable by Day 62 in both the IgG + NPAC and ADP + NPAC-treated groups (Figure 6A,B). While over time the tumors regrew within the primary region in the animals treated with IgG + NPAC, those in the ADP + NPAC-treated animals were robustly inhibited [ADP2 TGI = 82%, *p* = 0.027; and ADP13 TGI = 70%, *p* = 0.044] (Figure 6A,B). Furthermore, 30% of the ADP2 + NPAC-treated and 50% of the ADP13 + NPAC-treated mice displayed minimal residual disease (defined as a palpable mass of <10 mm^3^) at the end of the experiment. Thus, both ADP2 and ADP13 reduce TNBC locoregional regrowth following SoC NPAC treatment.

### 3.5. ADP2 or ADP13 Treatment Improves TNBC Outcome

The rapid primary tumor growth seen in our pre-existing orthotopic tumor model (Figure 3) does not allow enough time for metastases to become large enough so that they can be consistently detected using live or dissected organ imaging before the mice have to be sacrificed to comply with IACUC humane policies. Similarly, in our tumor regrowth model (Figure 6), the detection of metastases is hindered by their growth delay following CT treatment and the eventual (past 26 weeks of age), natural occurrence of unrelated lymphomas and death in the NOD/SCID strain. To overcome these challenges and assess the effects of ADP2 and ADP13 on metastasis and survival, a neoadjuvant treatment protocol, in which primary mammary tumors are surgically removed, was performed. Our previous studies indicate that CTCs can be detected in the blood of mice injected with MDA-MB-231 cells as early as 7 days post-MFP injection and that the extended timeline of a neoadjuvant model, in the absence of a primary tumor, leads to consistent, detectable metastases in over 85% of mice [6]. MDA-MB-231-luc cells, which preferentially metastasize to the bones, were injected in the MFP of 10-week-old female NOD/SCID mice and allowed to form tumors as above. Once tumors reached ~50–75 mm^3^ (2 weeks post-MFP injection), the mice were treated 3x/week with 10 mg/kg ADP2, ADP13, or their control IgGs. After 1 week of treatment, when the tumors in the control group had reached ~200 mm^3^ in volume, all the tumors were surgically removed, and the Ab treatment continued for an additional 12.5 weeks (88 days). Recurrence of a tumor at the primary site was detected using palpation, and tumor growth followed over time. The mice were sacrificed either when the recurrent tumors reached 0.9 cm^3^ (as death is not an acceptable endpoint in our IACUC protocol) or on Day 88 post-tumor resection (at 25.5 weeks of age) to avoid the occurrence of any spontaneous lymphomas that could confound the results. The days from surgery to detection of a palpable recurrence and from surgery to a humane or experimental endpoint were used to generate Kaplan–Meier curves for DFS and OS, respectively. Treatment with either ADP2 or ADP13 significantly increased both the DFS and OS (Figure 7A,B). Biophotonic imaging of the bones dissected at sacrifice revealed a profound reduction in metastasis following ADP2 (*p* = 0.038) or ADP13 (*p* = 0.034) treatment, as judged by decreased luciferase signal intensity (Figure 7C,D). Thus, ADP2 and ADP13 reduce metastasis and improve the survival of animals carrying aggressive TNBC tumors.

## 4. Discussion

Here, we report on the isolation of a new class of highly specific human ADAM8 mAbs that bind to sequences within its DI, inhibit both its MP and DI activities, and demonstrate substantial in vivo anti-cancer efficacy. These inhibitors were obtained using rHuADAM8 as an immunogen in mice to generate Ab-producing hybridomas coupled with a complex Ab screening approach. Abs were first selected for (i) strong binding to native ADAM8, (ii) lack of cross-reactivity to closely related ADAM proteins, and (iii) potent in vitro inhibition of both MP and DI activities in cell-based assays. Overall, a panel of mAbs (called ADPs) that fit these characteristics was isolated; these Abs were sorted into five epitope clusters (four overlapping and one unique) on ADAM8. Subsequent efficacy testing in an orthotopic mouse model of primary TNBC identified Epitope cluster 2 and 3 mAbs as an ADP subgroup with substantial in vivo activity. Flow cytometry, HDX-MS, and ALA scanning mutagenesis demonstrated that their function was mediated via binding to the DI (Figure 4 and Figure 5 and Appendix A). ADP2 and ADP13 were selected as the lead Abs based on a superior ability to inhibit not only primary TNBC tumor growth but also progressive disease. In a neoadjuvant tumor resection model, long-term treatment with either ADP2 or ADP13 demonstrated increased DFS, reduced metastasis, and improved OS. ADP2 and ADP13 also inhibited tumor locoregional regrowth following treatment with the TNBC SoC CT NPAC. Thus, our findings identify a family of highly specific ADAM8-neutralizing Abs and support their continued development into a future targeted treatment against ADAM8-driven TNBC.

To our knowledge, this is the first report of a mAb binding to the DI of an ADAM protein and inhibiting the activities of both its MP and DI. Because the ADAM8 protein has not been successfully crystallized in the absence of a pharmacologic inhibitor, it is difficult to identify the precise mechanism of inhibition. Given their localization of binding, however, it seems feasible that our lead ADP2 and ADP13 Abs disrupt the tight C-shaped DI/CRD 3D structure of ADAM8, thereby impacting the hypervariable (HVR) loop of the CRD that drives substrate recognition and cleavage by the MP necessary for activation of growth-promoting factors, while at the same time directly hindering the ability of the DI to interact with integrins and other ECM components required for metastasis. Of note, the clustering of some ADPs into a unique, non-overlapping epitope suggests dual inhibition may occur through several distinct mechanisms. Our ADAM8 dual MP and DI inhibitory Abs should have greater efficacy in the clinic compared to alternative approaches that have targeted only one domain. For example, substantial efforts have been made previously to inhibit the MP of ADAM8 using small molecules, despite demonstrated specificity and safety problems with this class of compounds [24]. The crystallization of ADAM8 in a complex with the broad-spectrum matrix metalloproteinase inhibitor Batimastat revealed the overall similarity of the ADAM8 3D MP structure to those of other ADAM family members, although it identified regions of deviation within the S1 specificity pocket of the catalytic domain, which could potentially be exploited for selective inhibition [12]. A more recent study, examining eight small-molecule MP inhibitors against ADAM8, similarly focused on the S1 pocket and demonstrated that ADAM8 and ADAM17 share active site geometry, placing into question the possibility of designing a truly selective small-molecule drug MP inhibitor [13]. Such an inhibitor, however, even if attainable, would likely fail to block DI activity, making it clinically inefficient in cancer therapy as tumor cells would be left able to invade tissues, infiltrate the blood stream, and establish metastases, the ultimate cause of patient mortality. The inhibition of human ADAM8 using a cyclic peptide that mimics a motif within the integrin-binding loop of ADAM8 has also been attempted. This inhibitor, BK-1361, designed to prevent multimerization and therefore autocatalytic activation, has the potential to fully inhibit protein function (both MP and DI activities) [14]. BK-1361 reduced pancreatic tumor burden and metastasis in mice, but a short half-life of only 30 min raises questions about its clinical value. A library of structural analogues based on systematic BK-1361 AA substitutions was subsequently generated but failed to improve its activity [15]. In contrast, preliminary pharmacokinetic, thermostability, and aggregation studies (not described here) suggest our Abs have a favorable therapeutic profile. This, combined with their target specificity and dual MP and DI inhibitory activity (to reduce both tumor growth and spread), makes them superior candidates for drug development.

TNBC is challenging to treat, in part, because it is histologically and molecularly heterogeneous [25], highlighting the need for more tailored treatments. Our earlier immunohistochemistry studies showed that approximately 34% (17/50 samples) of TNBC tumors are ADAM8+ at diagnosis and thus could potentially benefit from ADAM8 inhibition. Furthermore, ADAM8 mRNA was an independent predictor of a poor outcome in breast cancer patients [6]. More recently, we have developed an ADP2-based ADAM8 IHC assay, which confirmed the ADAM8 expression rate in TNBC (36.11% = 22/61 samples) and expanded ADAM8 positivity to all breast cancers, i.e., 33.9% (166 of 490 samples) of breast tumors of any subtype were ADAM8+ [7]. Furthermore, ER+/PR+/HER2- patients with high ADAM8 protein levels were at risk of poor survival in a 10-year age- and race-adjusted Cox proportional hazards model [7]. Recent studies highlight the need for identification and early intervention for breast cancer patients with aggressive tumors. For ADAM8+ TNBC patients, this could be of particular importance to mitigate disease progression. This approach is strongly supported by our neoadjuvant and tumor regrowth models with ADP2 and ADP13 (Figure 6 and Figure 7). Similarly, recent clinical evidence from the KEYNOTE-522 trial of pembrolizumab, which is now approved as SoC for high-risk early stage TNBC (defined as Stage II and III), demonstrated a significant benefit of such an early approach to avoid metastatic progression. Specifically, the addition of pembrolizumab to neoadjuvant CT improved the rate of pathological complete response (64.8% vs. 51.2%, *p* = 0.0005) and event-free survival (3-year: 84.5% vs. 76.8%, HR 0.63, and 95% CI 0.48–0.82) vs. neoadjuvant CT alone; the data on OS are still being collected [26,27]. Despite this positive result, a significant number of patients still had residual disease, highlighting the need for additional therapies to improve individual outcomes. Of note, ADP treatment of patient tumors is likely to result in greater efficacy than that seen in our orthotopic tumor mouse models, given the homogeneous staining of ADAM8 seen in patient breast cancer biopsies vs. TNBC MDA-MB-231 cell line-derived tumors, which have the limitation of having ADAM8 expression restricted to the tumor leading edge and areas of necrosis [6,7]. Ongoing studies seek to discover additional human breast cancer patient-derived xenograft or cell line models with uniform ADAM8 expression to demonstrate the full ADP therapeutic potential. In the future, studies will explore whether combinations of anti-ADAM8 Abs targeting distinct epitope regions, for example, ADP2 or ADP13 and non-overlapping Epitope 1 binding ADPs, or antibodies against the ADAM8 CRD or ELD could result in enhanced efficacy. Additionally, studies will focus on testing ADPs together with or compared against newly approved agents for TNBC treatment, including pembrolizumab and sacituzumab govitecan-hziy, which represent the most likely barriers to clinical entry for a future ADAM8-targeted therapy. Importantly, our data already demonstrate substantial efficacy for ADP2 and ADP13 in the neoadjuvant setting and in combination with CT, supporting their immediate transition into the next phase of therapeutic development. These mouse mAbs will next be humanized to reduce their risk of immunogenicity in future patients in a process that will simultaneously focus on retaining the high target specificity, affinity within the nM range (which is optimal for penetrance of therapeutic Abs within solid tumors), stability, and efficacy of ADPs. Notably, high ADAM8 expression has been detected in a variety of solid tumors in addition to breast, including colon, stomach, liver, pancreas, lung, head and neck, and bone, and in each cancer correlates with a poor outcome [6,28,29,30,31,32,33,34]. To date, there has been no successful targeted treatment against ADAM8-expressing cancers either marketed or, to our knowledge, in the pipeline. Knockout studies in mice have demonstrated that ADAM8-deficient animals develop normally and have a normal lifespan, suggesting this protein is non-essential under physiological conditions [35,36]. Consistently, our studies confirm its limited expression in normal human tissues ([6] and unpublished observations). Overall, these findings suggest that a future ADP-based ADAM8-targeted therapeutic regimen would represent a safe and effective treatment option.

## 5. Conclusions

These studies describe the isolation of a novel class of highly specific human ADAM8 dual MP and DI inhibitor mAbs and the identification of ADP2 and ADP13 mAbs as promising candidates for the future development of the first ADAM8-targeted cancer intervention. Such a therapy could revolutionize the treatment of patients with ADAM8-driven TNBC or another malignancy who currently face highly aggressive disease and inadequate treatment options.

## Figures and Tables

**Figure 1 pharmaceutics-16-00536-f001:**
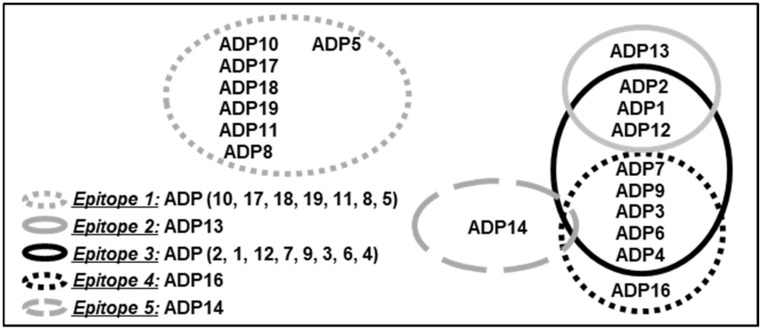
ADPs bind to five epitope clusters on ADAM8. Diagram indicating the epitope clusters for ADP binding on human ADAM8, and their partial overlap identified based on epitope binning using competitive ELISA.

**Figure 2 pharmaceutics-16-00536-f002:**
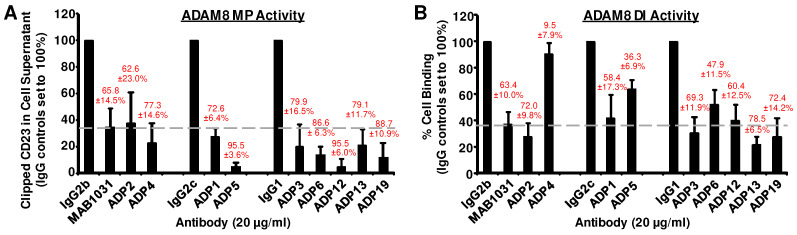
ADPs have potent in vitro dual metalloprotease and disintegrin (MP and DI) domain inhibitory activity. (**A**) ADAM8 MP activity was assessed in the presence of each ADP vs. its isotype-matched control IgG by measuring the release of soluble CD23 from the surface of HEK293 cells ectopically overexpressing both ADAM8 and CD23. After overnight antibody (Ab) treatment, conditioned cell media was tested for cleaved CD23 via the detection of its HA-tag in immunoblotting; images were quantified using densitometry. (**B**) ADAM8 DI activity was evaluated in the presence of each ADP vs. control IgG in assays measuring binding of α9β1-Integrin-expressing CHO cells to plates coated with recombinant human ADAM8 (rHuADAM8). Mean MP and DI activity level ± standard deviation (S.D.) from 3 independent experiments is graphed in A and B, respectively. The dashed line represents the level of activity in the presence of MAB1031. Ab-mediated percent inhibition of activity in each case was calculated as a decrease from control IgG levels, which were set to 100%. Mean percent inhibition for each ADP and for MAB1031 ± S.D. is given in red over the respective activity bars.

**Figure 3 pharmaceutics-16-00536-f003:**
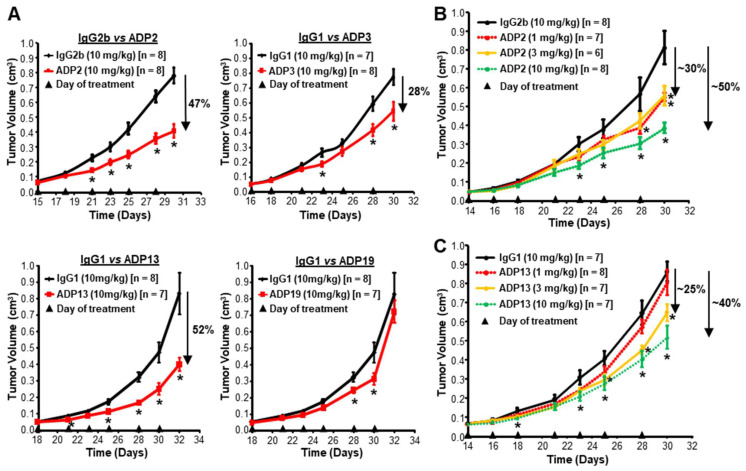
ADP2, ADP3, and ADP13 inhibit primary triple-negative breast cancer (TNBC) tumor growth in mice. (**A**) ADP2, ADP3, ADP13, and ADP19 were compared for their ability to inhibit pre-existing MDA-MB-231-luc TNBC cell line-derived orthotopic primary tumor growth. Female NOD/SCID mice carrying tumors of ~50–75 mm^3^ in volume were treated 3×/week with 10 mg/kg of the indicated ADP vs. their control IgG. Tumor growth was followed until tumors in the control group reached 1 cm^3^ (the limit of our IACUC protocol). Tumor volume [mean ± standard error of the mean (S.E.M.)] over time is presented. Percent tumor growth inhibition (TGI) for each ADP at the end of the experiment is given. *n* = number of animals/group. * *p* < 0.05 using a Student’s *t*-test. ADP2 (**B**) and ADP13 (**C**) were tested against pre-existing MDA-MB-231-luc-derived tumors as above in dose–response curves using 1, 3, 10, and 30 mg/kg vs. control IgG. Tumor volume (mean ± S.E.M.) over time is presented. Percent TGI at the end of the experiment is indicated. *n* = number of animals/group. *, *p* < 0.05 for treatment with indicated ADP dose vs. control IgG using a Student’s *t*-test. The 30 mg/kg dose of either ADP did not provide any additional benefit.

**Figure 4 pharmaceutics-16-00536-f004:**
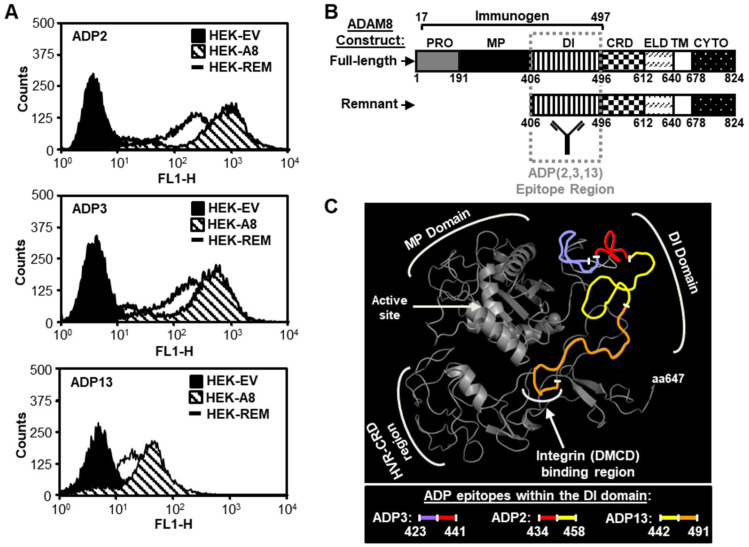
ADPs with in vivo anti-tumor activity bind to the ADAM8 DI. (**A**) ADP2, ADP3, or ADP13 binding to HEK293 cells expressing full-length ADAM8 (HEK-A8) vs. remnant form ADAM8 (HEK-REM) was assessed by flow cytometry; HEK293 cells expressing empty vector DNA (HEK-EV) were used as a negative control. Representative histograms of three independent runs are shown. (**B**) Schematic representation of the ADAM8 constructs used in part A, with domain information, amino acid (AA) numbers, and immunogen used for ADP generation indicated. The broad epitope region for ADP2, ADP3, and ADP13 binding to ADAM8, identified by the flow cytometry analysis in part A, is indicated (striped box). ADAM8 domains: PRO—prodomain; MP—metalloproteinase; DI—disintegrin; CRD—cysteine-rich; ELD—EGF-like; TM—transmembrane; CYTO—cytoplasmic. (**C**) Three-dimensional model of the predicted ADAM8 extracellular structure (residues 195-647, including MP, DI, CRD, and ELD) using the crystal structure of ADAM22 as template and Swiss-model software (2003). Regions of ADP2, ADP3, and ADP13 binding, including overlapping sequences, identified through hydrogen/deuterium exchange–mass spectrometry (HDX-MS) analysis are indicated. MP with active catalytic site, DI with integrin-binding region, and hypervariable region (HVR) of CDR are shown.

**Figure 5 pharmaceutics-16-00536-f005:**
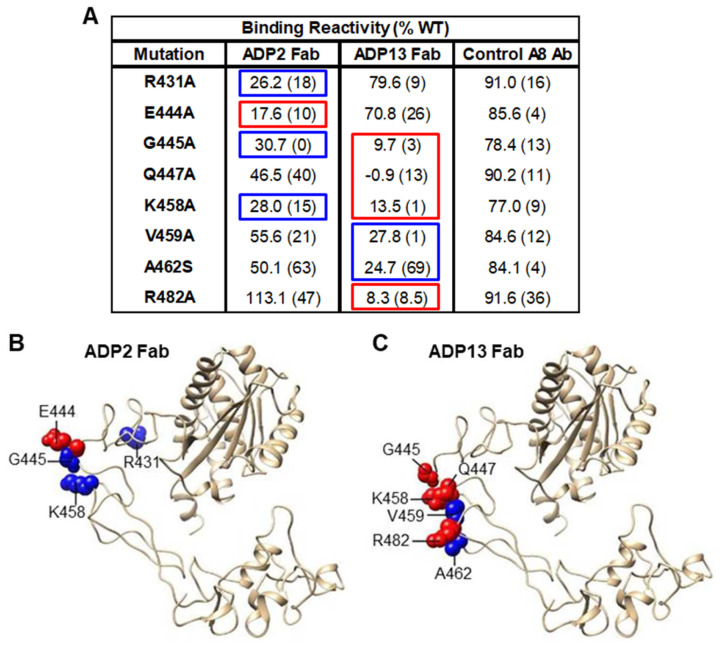
Amino acids (AAs) within the ADAM8 DI mediating ADP2 and ADP13 binding. (**A**) AA residues important for ADP2 and ADP13 binding to ADAM8 were identified using alanine (ALA) scanning mutagenesis plus flow cytometry. Mean binding reactivity (in duplicate samples) of ADP2 or ADP13 antigen−binding fragments (Fabs) to ADAM8 protein mutated at the indicated residues (mutation) within the MP and DI vs. binding of a positive control ADAM8 Ab (Control A8 Ab), which binds outside the MP and DI regions and is therefore unaffected, is presented as a percentage of binding to wild−type (WT) ADAM8. The range of binding reactivity (maximum–minimum) in each case is indicated in parentheses. AAs identified as critical for binding (i.e., those for which Control A8 Ab binding was >70% of WT but test Ab binding was <20% of WT binding) are shown in red boxes. Blue boxes show residues of secondary importance, i.e., AAs in close proximity to critical residues whose mutation led to a substantial (although not <20% of WT) reduction in binding. Epitope AAs for ADP2 (**B**) and ADP13 (**C**) Fab binding, identified through mutagenesis, are indicated on a crystal structure model of the ADAM8 ectodomain based on the structure of vascular apoptosis−inducing protein−1.

**Figure 6 pharmaceutics-16-00536-f006:**
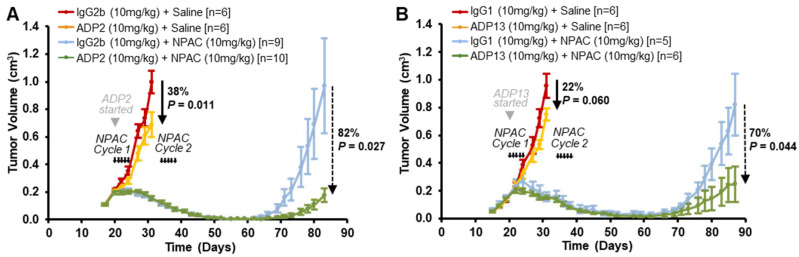
ADP2 and ADP13 inhibit TNBC tumor regrowth following standard-of-care (SoC) Nanoparticle Albumin-Bound Paclitaxel (NPAC) treatment. Female NOD/SCID mice were injected in the mammary fat pad (MFP) with MDA-MB-231-luc TNBC cells. When tumors reached ~150 mm^3^, mice were distributed into groups and treated with either (**A**) IgG2b + Saline, ADP2 + Saline, IgG2b + NPAC, or ADP2 + NPAC and (**B**) IgG1 + Saline, ADP13 + Saline, IgG1 + NPAC, or ADP13 + NPAC. NPAC was administered at 10 mg/kg in 2 cycles of 5 consecutive intravenous (i.v.) daily treatments (5 short arrows) with one week of rest in between; an equivalent volume of vehicle Saline was also given. ADP or control IgG was given by intraperitoneal (i.p.) injection, starting with a loading dose of 20 mg/kg on the day of first NPAC injection, followed by maintenance doses of 10 mg/kg 3×/week thereafter. Tumor volume was measured 3×/week, and mean ± S.E.M. is shown over time. Following regression, tumor recurrence was detected by palpation and growth (tumor volume) followed as above. Comparison of TGI ended when average tumor volume in the appropriate control groups reached 1 cm^3^. *p* values obtained using a Student’s *t*-test are indicated. *n* = number of animals/group.

**Figure 7 pharmaceutics-16-00536-f007:**
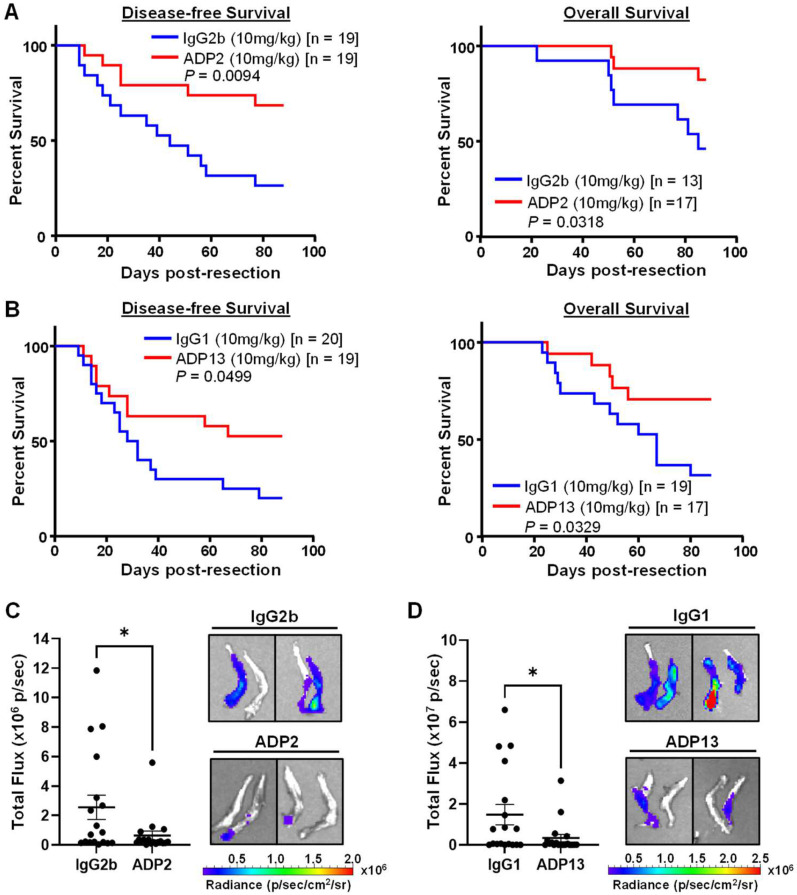
Neoadjuvant treatment with ADP2 or ADP13 improves outcome in mice. When MFP tumors derived from MDA-MB-231-luc TNBC cells reached a volume of ~50–75 mm^3^, mice were treated 3× with 10 mg/kg of either ADP2, ADP13, or their isotype-matched control IgGs. At a volume of ~200 mm^3^, tumors were surgically removed, and i.p. Ab treatment continued 3×/week for an additional 12.5 weeks. Recurrence of a tumor at the primary site was detected using palpation. Tumor size was measured 3×/week and volume calculated. Mice were sacrificed when recurrent tumors reached 0.9 cm^3^, as death is not an acceptable endpoint in our IACUC protocol, or at the end of the experiment (Day 88 post-tumor resection). Kaplan–Meier curves for disease-free survival (DFS) and overall survival (OS) of ADP2 (**A**) and ADP13 (**B**) vs. control IgG-treated animals are presented. *p* values were obtained using a Log rank test and are as indicated; *n* = number of mice per group. Bones from mice treated with ADP2 (**C**) and ADP13 (**D**) were dissected at sacrifice and examined for metastasis using Biophotonic imaging for detection of the activity of the luciferase tag expressed in MDA-MB-231-luc TNBC cells. Left panels: Luciferase activity (total flux in photons per second [p/s]) for individual mice in each treatment group is presented. Mean ± S.E.M. for each group is given. For IgG2b and ADP2: *n* = 18/group; for IgG1 and ADP13, n was 18 and 19, respectively. *p* values (ADP2: *p* = 0.038, and ADP13: *p* = 0.034) were obtained using an unpaired *t*-test. *, statistically significant, *p* < 0.05; Right panels: Representative images of hind leg bone metastases are shown.

**Table 1 pharmaceutics-16-00536-t001:** ADPs bind to native ADAM8. Flow cytometry was performed using HEK293 cells expressing full-length ADAM8 (HEK-A8) vs. an empty vector control DNA (HEK-EV) and decreasing antibody (Ab) concentrations. Mean Fluorescent Intensity (MFI) indicates extent of binding of each Ab.

Binding to Native ADAM8 in FACS (MFI)						
Cell Line	HEK-A8			HEK-EV		
**Ab (µg/mL)**	**10**	**1**	**0.1**	**10**	**1**	**0.1**
ADP1	185.7	171.2	46.8	3.1	3.1	3.1
ADP2	158.1	156.5	58.4	2.6	2.7	2.9
ADP3	86.9	96.5	51.8	2.8	2.7	2.7
ADP4	160.8	140.0	47.4	2.6	2.7	2.7
ADP5	149.0	105.9	26.8	2.7	2.8	3.1
ADP6	62.4	65.4	28.2	2.7	2.7	2.9
ADP7	68.1	63.1	25.8	2.7	2.9	2.8
ADP8	48.6	53.1	17.0	2.6	2.6	2.6
ADP9	62.1	61.1	22.1	3.2	2.9	2.7
ADP10	67.2	58.2	18.7	3.2	2.8	2.7
ADP11	56.6	64.2	24.1	3.1	2.7	2.6
ADP12	106.7	68.7	17.4	3.1	3.1	3.0
ADP13	62.2	21.1	7.1	2.6	2.7	2.6
ADP14	58.6	17.9	5.9	3.3	3.2	3.0
ADP16	147.3	89.6	24.6	2.6	2.9	2.9
ADP17	53.6	63.8	25.5	2.8	2.7	2.8
ADP18	53.7	61.0	23.5	2.8	2.8	2.7
ADP19	59.0	52.9	17.5	2.6	2.6	3.0

**Table 2 pharmaceutics-16-00536-t002:** ADPs bind with high affinity to recombinant human ADAM8 (rHuADAM8). The half-maximal effective concentration (EC50) obtained in ELISA and binding kinetics (k_a_, k_d_, and K_D_) obtained through Biacore studies for each ADP are shown.

Binding to rHuADAM8				
	ELISA	Biacore		
**Ab**	**EC50 (nM)**	**k_a_ (1/Ms)**	**k_d_ (1/s)**	**K_D_ (M)**
ADP1	0.032	7.04 × 10^4^	3.15 × 10^−4^	4.47 × 10^−9^
ADP2	0.029	8.95 × 10^4^	2.99 × 10^−4^	3.34 × 10^−9^
ADP3	0.049	3.47 × 10^4^	6.34 × 10^−4^	1.83 × 10^−8^
ADP4	0.041	3.52 × 10^4^	2.55 × 10^−2^	7.23 × 10^−8^
ADP5	0.059	9.51 × 10^3^	9.27 × 10^−5^	9.75 × 10^−9^
ADP6	0.059	2.62 × 10^5^	4.07 × 10^−3^	1.55 × 10^−8^
ADP7	0.091	1.68 × 10^4^	8.89 × 10^−4^	5.29 × 10^−8^
ADP8	0.097	3.14 × 10^3^	1.48 × 10^−4^	4.72 × 10^−8^
ADP9	0.053	2.78 × 10^4^	1.37 × 10^−3^	4.92 × 10^−8^
ADP10	0.091	2.45 × 10^3^	7.52 × 10^−5^	3.07 × 10^−8^
ADP11	0.097	5.07 × 10^3^	1.09 × 10^−4^	2.14 × 10^−8^
ADP12	0.061	4.97 × 10^4^	4.14 × 10^−4^	8.33 × 10^−9^
ADP13	0.054	3.54 × 10^4^	4.60 × 10^−5^	1.30 × 10^−9^
ADP14	0.531	2.32 × 10^4^	6.12 × 10^−4^	2.64 × 10^−8^
ADP16	0.053	7.27 × 10^4^	5.12 × 10^−3^	7.05 × 10^−8^
ADP17	0.049	6.78 × 10^3^	1.26 × 10^−4^	1.86 × 10^−8^
ADP18	0.024	8.05 × 10^3^	1.23 × 10^−4^	1.53 × 10^−8^
ADP19	0.085	1.60 × 10^4^	1.45 × 10^−4^	9.11 × 10^−9^

## Data Availability

All data generated or analyzed during this study are included in this article and its Appendix A.

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
