# Peer review of "A Novel Class of Human ADAM8 Inhibitory Antibodies for Treatment of Triple-Negative Breast Cancer"

_pharmaceutics, 2024, doi:10.3390/pharmaceutics16040536_

Round 1

Reviewer 1 Report

Comments and Suggestions for Authors

In this manuscript, the authors present evidence of the efficacy of two monoclonal antibodies (mAbs) that simultaneously inhibits two domains of the cell surface protein A Disintegrin And Metalloprotease 8 (ADAM8) as potential therapeutic agents for Triple-negative breast cancer (TNBC).

The authors describe the process of generating a hybridoma library using recombinant human ADAM8, followed by screening to isolate a novel panel of highly specific human ADAM8 inhibitory mAbs, termed ADPs, through flow cytometry with HEK-A8 cells and ELISA plates coated with rHuADAM8. They excluded cross-reactive clones that reacted with ADAM9, ADAM12, or ADAM15, resulting in a panel of 19 ADPs.

The affinity of the ADP panel to rHuADAM8 was assessed using ELISA, revealing an EC50 ranging between 0.024 and 0.531 nM, and using Biacore, yielding a Kd between 1.30E-09 and 7.23E-08. Next, epitope binning analysis identified the ADPs´ binding to five different epitope clusters. The top nine ADPs exhibiting higher affinity were selected to test the metalloproteinase (MP) and disintegrin (DI) inhibitory activity ADAM8 dependent, with eight displaying potent dual MP and DI inhibitory activity in vitro.

Later, the authors determined that ADP2, ADP3, and ADP13 exhibit significant therapeutic efficacy in vivo in female NOD/SCID mice with an orthotopic MDA-MB-231 TNBC model of ADAM8-driven primary tumor growth.  

The authors mapped the binding site of ADP2, ADP3, and ADP13 on ADAM-8 using flow cytometry with ADAM-8 constructs exhibiting deleted regions, facilitating the identification of the epitope regions. Further analysis using HDX-MS enabled epitope mapping at the peptide level for ADP2 (residues 434-458), ADP3 (residues 423-441), and ADP13 (residues 442-491). They could observe a significant epitope overlap between ADP2 and ADP3, between ADP2 and ADP13, but not between ADP3 and ADP13.  Moreover, the authors performed Alanine scanning mutagenesis in ADAM8 to identify residues in the DI region crucial for binding with functional activity for ADP2 and ADP13, demostrating high concordance with the HDX-MS analysis.

Finally, ADP2 and ADP13 were assessed to evaluate their effect on tumor regrowth following standard-of-care (SoC) Nanoparticle Albumin-Bound Paclitaxel (NPAC) treatment in an aggressive TNBC tumor model using NOD/SCID mice bearing advanced, rapidly growing MDA-MB-231-luc cell-derived mammary tumors. Notably, ADP2 and ADP13 significantly inhibited TNBC tumor regrowth and significantly improved the outcome in a neoadjuvant treatment model.

Overall, the article is well written, with an appropriate research design. The authors describe properly each step in the development of the ADPs, including the affinity determination, in vitro activity, epitope mapping, and in vivo activity employing relevant models. The authors provide here compelling evidence that ADP2 and ADP13 are promising candidates for development of ADAM-8 targeted therapies and the article should be accepted for publication.

Reviewer 2 Report

Comments and Suggestions for Authors

In the present manuscript, the authors present the isolation and characterization of hybridoma antibodies targeting ADAM8, an important factor in tumor spread and growth of TNBC and other cancer types. Using competition assays, different hits were mapped to five different, partially overlapping, epitope bins. Their favuorite antibody candidates inhibit both metalloproteinase and disintegrin activity, and the two lead molecules show anti-cancer activity in vivo in an orthotopic TNBC mouse model with MDA-MB231 cells. For both of these antibodies, they mapped the epitope of binding to the disintegrin domain, via a spectrum of orthogonal methods including FACS, HDX-mass spectrometry and alanine scanning mutagenesis of the antigen.  

This is a very valuable contribution with the potential to positively impact the opinion on ADAM8 as a therapeutic target. The spectrum of methods used and the amount of data shown is very impressive, still the article is very clear and easy to read. My only suggestion to the authors would be to modify the discussion: as quite clearly, they propose the therapeutic application of isolated antibodies, comments on which further development steps are envisioned would be welcome as an outlook. In particular: as these are primary hits, is the affinity sufficient or do they expect higher affinity Abs to work better? Would combination therapy (either with oligoclonals or small molecules) improve the anti-tumor effect, and how could the mix be diesigned? Should the isolated candidates be humanized or are they consider “surrogates” and will fully human antibodies be isolated?

Please find below a list of minor remarks which I hope you will find helpful.

Material and methods section: please use the RRIDs also for commercially acquired antibodies, where possible

Line 161: 1.25 μg, surely you wanted to state the concentration?

Line 172: molarity of glycine buffer?

Line 219: CRD-ELD abbreviation not explained

Line 314: unit (M) is missing

From line 323: are Figure 2 and Table 3 presenting the same data? In this case I would propose to merge them into one Figure, and labelling the percent inhibition with the deviation over the respective bars.

Line 454: the usual citation of PDB structures is: PDB: 2ERP, for example
